# Survey of metaproteomics software tools for functional microbiome analysis

Ray Sajulga[1], Caleb Easterly[1], Michael Riffle[2], Bart Mesuere[3], Thilo Muth[4], Subina Mehta[1], Praveen Kumar[1], James Johnson[1], Bjoern Andreas Gruening[5], Henning Schiebenhoefer[6], Carolin A. Kolmeder[7], Stephan Fuchs[8], Brook L. Nunn[2], Joel Rudney[1], Timothy J. Griffin[1], Pratik D. Jagtap[1]*

1 University of Minnesota, Minneapolis, Minnesota, United States of America, 2 University of Washington, Seattle, Washington, United States of America, 3 Ghent University, Ghent, Belgium, 4 Federal Institute for Materials Research and Testing, Berlin, Germany, 5 University of Freiburg, Freiburg, Germany, 6 Robert Koch Institute, Berlin, Germany, 7 University of Helsinki, Helsinki, Finland, 8 Robert Koch Institute, Wernigerode, Germany

* pjagtap@umn.edu

**Data Availability Statement:** The data underlying the results presented in the study are available from https://zenodo.org/record/4067104; DOI: 10. 5281/zenodo.4067104.

## Abstract

To gain a thorough appreciation of microbiome dynamics, researchers characterize the functional relevance of expressed microbial genes or proteins. This can be accomplished through metaproteomics, which characterizes the protein expression of microbiomes. Several software tools exist for analyzing microbiomes at the functional level by measuring their combined proteome-level response to environmental perturbations. In this survey, we explore the performance of six available tools, to enable researchers to make informed decisions regarding software choice based on their research goals. Tandem mass spectrometry-based proteomic data obtained from dental caries plaque samples grown with and without sucrose in paired biofilm reactors were used as representative data for this evaluation. Microbial peptides from one sample pair were identified by the X! tandem search algorithm via SearchGUI and subjected to functional analysis using software tools including eggNOG-mapper, MEGAN5, MetaGOmics, MetaProteomeAnalyzer (MPA), ProPHAnE, and Unipept to generate functional annotation through Gene Ontology (GO) terms. Among these software tools, notable differences in functional annotation were detected after comparing differentially expressed protein functional groups. Based on the generated GO terms of these tools we performed a peptide-level comparison to evaluate the quality of their functional annotations. A BLAST analysis against the NCBI non-redundant database revealed that the sensitivity and specificity of functional annotation varied between tools. For example, eggNOG-mapper mapped to the most number of GO terms, while Unipept generated more accurate GO terms. Based on our evaluation, metaproteomics researchers can choose the software according to their analytical needs and developers can use the resulting feedback to further optimize their algorithms. To make more of these tools accessible via scalable metaproteomics workflows, eggNOG-mapper and Unipept 4.0 were incorporated into the Galaxy platform.

**Funding:** T.G. National Cancer Institute - Informatics Technology for Cancer Research (NCI-ITCR) grant 1U24CA199347 and National Science Foundation (U.S.) grant 1458524 P.D.J: Extreme Science and Engineering Discovery Environment (XSEDE) research allocation BIO170096 B.G: Collaborative Research Centre 992 Medical Epigenetics (DFG grant SFB 992/1 2012) and German Federal Ministry of Education and Research (BMBF grants 031 A538A/A538C RBC, 031L0101B/031L0101C de.NBI-epi, 031L0106 de. STAIR (de.NBI)). The funders had no role in study design, data collection and analysis, decision to publish, or preparation of the manuscript.

**Competing interests:** The authors have declared that no competing interests exist.

## Introduction

Microbiome research has demonstrated the effect of microbiota on their host and environment [1,2]. To determine the key contributors within complex microbiota, nucleic acid-based methods such as metagenomics can identify taxonomies that are prevalent in certain environments and stimuli [3]. Metagenomics provides an overview of the complete inventory of genes recovered from microbiome samples. As a gene-centric approach it is static and therefore cannot fully reflect the temporal dynamics and functional activities of microbiomes. To gain a more impactful understanding of a microbiome, metaproteomics must be used to determine the actions, or functions, of microbial organisms. Specifically, metaproteomics identifies microbial proteins, which are biological units of function [4,5]. Functional analysis also helps in understanding the mechanism by which microorganisms interact with each other and their immediate environment, thus offering deeper insights beyond mere taxonomic composition and correlation of the microbiome [6]. For example, functional analysis can provide information about which enzymes are active in particular biological processes and metabolic pathways. Thus, investigating the functional metaproteome aims to give biological relevance to the structure of microbiomes and can help to identify metabolism changes caused by specific perturbations and environmental factors.

Metaproteomics data analysis involves primarily identification of peptides from tandem mass spectrometry (MS/MS) data by matching it against protein sequence databases. The identified peptides are assigned to proteins or protein groups, some of which are unique while others are shared amongst various taxa. These identifications are further assigned to functional groups using various annotation databases [7]. Compared to single-organism proteomics, the functional annotation of metaproteomics data is not straightforward because it involves multiple complex layers: identified peptides can be assigned to various sequence-similar proteins originating from multiple organisms. This adds on to the already existing challenge of assigning functional groups because proteins can often be assigned to multiple functional groups.

Over the years, multiple software tools have been developed to assess the functional state of a microbial community based on the predicted functions of proteins identified by MS [8–13]. These tools differ in various aspects such as–a) input files used for processing, b) annotation databases used for protein and functional assignment, c) peptide- versus protein-level analysis, d) generated outputs including functional ontology terms, and e) visual outputs generated for biological interpretation (Fig 1). Each functional tool has its advantages, and labs across the world have been using them often based on the criterion of how well a certain tool fits into their bioinformatics workflow. However, to our best knowledge, these functional tools have not been compared on the same biological dataset in any benchmarking study yet [14].

In this study, we compared and evaluated the performance of six open-source software tools—eggNOG-mapper [8], MEGAN [9], MetaGOmics [10], MetaProteomeAnalyzer (MPA) [11], ProPHAnE (https://www.prophane.de) and Unipept [13]—that specialize in performing functional analysis of metaproteomics data. For this purpose, we used a published oral dysbiosis dataset [15] to generate raw outputs and compare features such as identification statistics, functional group assignment (both at a dataset and individual-peptide level), and quantitative analysis features such as differential protein expression. We observed significant variability in results from the different functional tools. For a fairer comparison, we also expanded the Gene Ontology (GO) terms using metaQuantome software [16] and compared functional outputs. metaQuantome is a software tool that performs quantitative statistics on peptide-level quantitative, functional and taxonomic data to generate quantitative outputs for taxonomy, functions and taxonomy-function interaction. Based on this investigation, we provide some insights on the sources of this variability and offer suggestions on the usage of these functional tools.

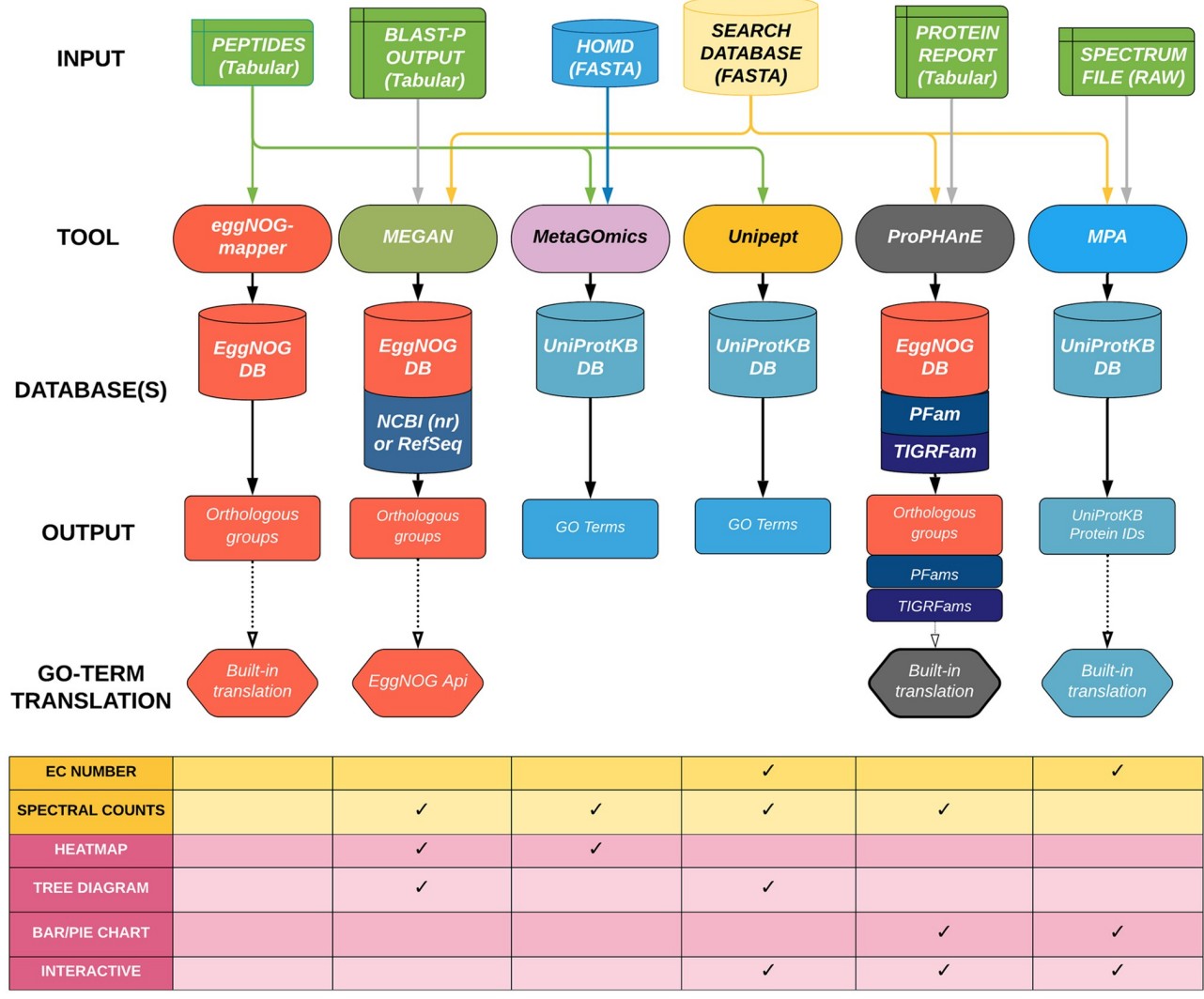

| | | | | | | |
|---|---|---|---|---|---|---|
| **EC NUMBER** | | | | ✓ | | ✓ |
| **SPECTRAL COUNTS** | | ✓ | ✓ | ✓ | ✓ | |
| **HEATMAP** | | ✓ | ✓ | | | |
| **TREE DIAGRAM** | | ✓ | | | ✓ | |
| **BAR/PIE CHART** | | | | | ✓ | ✓ |
| **INTERACTIVE** | | | | ✓ | ✓ | ✓ |

**Fig 1. A comparative workflow of all six functional software tools producing GO term lists from the same dataset.** The inputs that are required for each software tool are connected from the top. The reference databases used for each tool are aligned in the middle. The outputs and Gene Ontology (GO) term translation processes are outlined at the bottom. Additional output types (data and visualizations) are shown in the table underneath the workflow.

We have packaged two of these software tools, Unipept 4.0 and eggNOG-mapper, into the Galaxy platform [17], based both on their performance and amenability to deployment in a workflow engine like Galaxy. This will make these tools widely accessible to users and facilitate their usage in analytical workflows.

## Methods

### Input dataset and search results

The basis for this study lies in selecting a dataset suitable for metaproteomics comparison. Here, we used a published oral dysbiosis dataset [15] containing mass-spectrometry data generated from plaque extracted from dental caries-prone children. This data is publicly available in the ProteomeXchange Consortium (http://proteomecentral.proteomexchange.org) via the

PRIDE partner repository with dataset identifier PXD003151. These plaques were incubated within biofilm reactors for 72 hours and subject to a pair of conditions for the last 48 hours: with sucrose-pulsing (WS) five times a day and with no sucrose (NS) but a constant basal mucin medium flow. One sample pair (737 NS/WS) was selected since they were sufficiently separated based on principal component analysis [15]. Although not a 'ground-truth' dataset, this dataset was chosen since it was previously analyzed for its biological significance [15,16]. From the 737 dataset, Mascot Generic Format (MGF) files containing peak lists were used to identify peptides using X! Tandem search algorithm via SearchGUI. Each of these peptide-to-spectrum matches (PSMs) represent a specific peptide sequence mapped to a specific spectrum scan, as well as match information such as retention time, measured mass-to-charge ratio, theoretical mass and charge, and confidence score. To determine which proteins these identified peptides belong to, a combined protein database was provided: the Human Oral Microbiome Database (HOMD—1,079,742 protein sequences—May 2017) concatenated with the common Repository of Adventitious Proteins (cRAP). Using cRAP proteins to filter out contaminants, X! Tandem identified ~27,000 microbial peptides from each dataset (Supplement Data: https://zenodo.org/record/4067104; DOI: 10.5281/zenodo.4067104). Regarding specific SearchGUI parameters, peptides were assigned based on trypsin-digested proteins, with up to two missed cleavage sites allowed. Amino acid modifications were specified as methylthio of cysteine (fixed modification) and oxidation of methionine (variable modification). The accepted precursor mass tolerance was set to 10 ppm and the fragment mass tolerance to 0.05 daltons with a charge range from +2 to +6. To interpret SearchGUI results, PeptideShaker was used to generate a PSM, peptide, and protein report, with a false-discovery rate (FDR) cutoff at 1%. Only peptides with length from 6 to 30 amino acids were considered. Spectral counts were calculated based on the number of PSMs assigned for each peptide. Peptide search results were processed to provide appropriate inputs for each of the metaproteomics software tools. (Supplement Data: https://zenodo.org/record/4067104; DOI: 10.5281/zenodo.4067104).

## Functional tools

Six functional analysis tools were used to analyze the data: eggNOG-mapper (version 1.0.3), MEGAN5, MetaGOmics (version 0.1.1), MPA (version 1.8.0), Unipept 4.0, and ProPHAnE (3.1.4). Standard procedures were used for each tool, as defined by their developers. Input types and databases are specified in Fig 1.

For eggNOG-mapper, we used the Galaxy-implemented version of this tool (v1.0.3) on our local Galaxy for Proteomics (Galaxy-P) server. DIAMOND (Double Index Alignment of Next-generation sequencing Data) was used as a mapping mode. PAM30 (Point Accepted Mutation) was used as a scoring matrix (gap costs with an existence value of 9 and extension value of 1). Bacteria were used for taxonomic scope, and all orthologues were considered. Gene ontology evidence was based on non-electronically curated terms, and seed orthology search options had a minimum e-value of 200,000 and a minimum bit score of 20.

For MetaGOmics, a list of peptides and the HOMD were uploaded (version 0.2.0; https://www.yeastrc.org/metagomics). Searches were performed using the UniProt/SwissProt database with a BLAST e-value cutoff at 1e-10, using only the top hit.

In MEGAN, for Lowest Common Ancestor (LCA) analysis, we used a minimum score of 30 and a maximum expected threshold value of 3.0. Hits with a BLAST score in the top 10% were chosen for further processing. The minimum support filter to reduce the false positive hits was set as five. Naive LCA algorithm was used with 100% coverage. The read assignment mode was set to read counts. The read-match-archive (RMA) files were generated and are shared via https://zenodo.org/record/4067104; DOI: 10.5281/zenodo.4067104. Further details

on how the RMA files were generated from MEGAN5 have been described in the dysbiosis study [15]. Functional analysis was performed to map reads to genes with known COGs (clusters of orthologous groups) and NOGs (non-supervised orthologous groups). These groups were then translated to GO terms using the following eggNOG RESTful API: http://eggnogapi5.embl.de/nog_data/json/go_terms/.

MPA performs its own searches using X! Tandem, rather than using the output from SearchGUI. Specifically, MPA takes in the same input MGF files for SearchGUI (MGF files). In order to enable searches, a FASTA database generated using the sectioning method [18] was used. To maintain comparable results, the same parameters as for SearchGUI were used for the X! Tandem search within MPA: precursor mass tolerance was set at 10 ppm and fragment mass tolerance was set at 0.05 daltons. Two missed cleavages were allowed and trypsin was used for protein digestion. The false discovery rate (FDR) was set at 1%. For grouping proteins, a minimum of one shared peptide was used if relevant to the analysis.

Unipept takes in a tabular list of peptides. Search settings were configured for isoleucine and leucine to be considered distinct, duplicate peptides to be retained, and advanced missed cleavage handling to be enabled.

In ProPHAnE, the NS and WS samples were grouped into one sample group each. Metaproteins (clustered by MPA) were analyzed using default parameters. Functional annotation was transferred from TIGRfams v15 and Pfams v32 using HMMScan with trusted cut-off and, additionally, EggNOG v4.5.1 using eggNOG-mapper (hmmer mode).

## GO term standardization

The six tools described here produce different functional annotation types. To compare and contrast the composition of each tool's functional annotation, a standard annotation type was chosen. Gene Ontology (GO) terms were used since they are a well-supported and common annotation type throughout most of the tools through either native outputs (e.g., MetaGOmics and Unipept) or in-built translations (e.g., MEGAN, ProPHAnE, MPA) and are commonly used in the metaproteomics community. However, MEGAN, ProPHAnE, and MPA do not provide direct GO term outputs and thus require external databases for translation. MEGAN and ProPHAnE produce eggNOG orthologous group accessions, which are translated using eggNOG's API (http://eggnogdb.embl.de/#/app/api), while MPA produces UniProt protein accessions, which are used to retrieve UniProtKB GO terms via UniProt's Retrieve/ID mapping tool (https://www.uniprot.org/uploadlists/).

When using GO terms, it is important to consider their categorization into three different domains: molecular function, biological process, and cellular component. Molecular functions describe the biological activity of gene products at a molecular level (e.g., ATPase activity), biological processes represent widely encompassing pathways that can involve many proteins that aim to accomplish specific biological objectives (e.g., regulation of ATPase activity), and cellular components describe the localization of the activity of these gene products (e.g., plasma membrane). To achieve a high-resolution analysis on functional annotation, only molecular function GO terms were the focus of this analysis. However, since it is important to consider how each tool handles the other ontologies as well, analyses using biological processes, cellular components, and all three domains combined are provided in the supplementary section.

## Overlap analysis

As an initial evaluation, the GO term outputs of each tool were compared via an R markdown notebook (https://github.com/galaxyproteomics/functional-analysis-benchmarking/blob/master/func_tools_analysis.Rmd) that looked at the total number of GO terms and the total

number of unique terms for each tool. Next, the degree of dissimilarity between the GO term output sets was gauged using fractional overlap indices calculated for each tool pairing. These values were calculated for a tool by taking the size of its intersected GO set with another tool and dividing by the size of the original tool's term set. To examine the relationships between more than two tools, a six-set Venn diagram was constructed through InteractiVenn [19].

To gain a better understanding of how each tool's GO output set overlaps with each other, a single GO term was selected for in-depth analysis. Two criteria for selecting the GO term were used to ensure a comprehensive yet focused comparison: (1) presence in all six tools and (2) minimal presence within each tool. Once selected, any GO terms associated with that selected term were extracted for overlap analysis between the tools. The associated GO terms were then evaluated to determine any hierarchical relationships. GO terms assume a hierarchical structure (directed acyclic graphs) which can be navigated using labels such as ancestor and descendant. For any GO term, its ancestors may consist of any number of less-specialized terms in its hierarchy. Conversely, its descendants consist of more-specific terms that would continue beneath its hierarchy [20]. For example, we can examine the ancestors and descendants of ion binding (GO:0043167): *metal ion binding* (descendant) → *cation binding* (child) → *ion binding* → *binding* (parent) → *molecular function* (ancestor).

## Expansion and filtering

The hierarchical nature of GO terms can vary greatly from tool to tool. To mitigate this factor in this comparative overlap analysis, we used an existing tool, metaQuantome [16], to expand the hierarchies for all GO terms in each functional tool. metaQuantome originally exists as a versatile metaproteomics software suite for aggregating peptide-level quantitative data for both taxonomy and function. One of its modules, metaQuantome expand, can be used to infer the presence of all possible ancestor GO terms. Additionally, quantitative information can be aggregated from multiple descendants of an ancestor [16,21]. By expanding these hierarchies, any fundamental differences between tools are more pronounced. For this analysis, the GO term sets for all tools were expanded and compared again using the previous fractional overlap analysis used for the full sets. Any datasets that were too large and were found to consume more than 1TB of RAM were collapsed by grouping via GO term and adding their intensities.

Once expanded, these GO hierarchies can contain extraneous or unsupported information. To curtail this issue, these expanded GO terms (ancestors) can be filtered out based on how many original GO terms (descendants) they originated from [16]. In this study, filtering was set to remove expanded GO terms that had fewer than two connections to original GO terms.

## Quantitative comparative analysis

After comparing the GO term outputs from each tool, fold changes for each term were evaluated (see below). This type of comparison is important in the analysis of microbiomes, since this can reveal which biological functions show an increase or decrease in abundance in response to a stimulus. For our analysis, peptide spectral counts associated with each GO term for both WS and NS conditions were used to estimate fold changes. These counts were available in the peptide reports from PeptideShaker. To ensure fair comparison, spectral count normalization between the two conditions was performed ad hoc. For eggNOG-mapper, MEGAN, and Unipept, a PSM fraction was used to scale down the condition with more spectral counts (NS) to match the other condition (WS): $\frac{Sum\ of\ WS\ PSMs}{Sum\ of\ NS\ PSMs}$. For MPA, we used protein-based normalization, wherein a protein fraction was used to normalize the GO term spectral counts before calculating the log ratio: $\frac{Sum\ of\ WS\ protein\ spectral\ counts}{Sum\ of\ NS\ protein\ spectral\ counts}$. MetaGOmics and ProPHAnE

already internally normalize their values through a normalized spectral abundance factor (NSAF) normalization method [22]. Once normalized, these quantitative values were used to calculate the fold changes as $log_2(\frac{WS+1}{NS+1})$. Using these fold changes, the GO terms were sorted in a descending order. Thus, GO terms that were found to be more abundant in WS conditions were found at the top of these rankings.

Two levels of comparisons of these fold changes were used: (1) between tools that natively output GO terms (i.e., Unipept and MetaGOmics); and (2) between all tools, which includes translated GO terms. Initially, Unipept and MetaGOmics were compared by taking the GO terms common between them and plotting each term based on the fold change calculations for Unipept (x-coordinate) and MetaGOmics (y-coordinate). A linear model was regressed, and the Pearson correlation coefficient was calculated with a two-sided alternative hypothesis. Secondly, all tools were included in a fold change comparison with translated GO terms. Using the functional annotation outputs from each tool, fold changes were estimated for each molecular function GO term and compared with one another. For each tool, the top five upregulated and top five downregulated terms were calculated with their fold changes and compared with the other tools. If one of these top terms was present in another tool's GO set, then that other tool's GO fold change calculation was referenced for comparison. If multiple copies exist in another tool, then the GO term with the largest absolute fold change was referenced. For primary analysis, a tool that aggregated GOs intrinsically (MetaGOmics or Unipept) with high coverage was selected. Similar analyses for the other tools are included in the supplement (**Supplement S4 Tables 3–7** in S1 File).

## Single peptide analysis

With an overall sense of tool discrepancies through identification and quantitation comparisons, the underlying differences between tools were closely examined through a single-peptide analysis. Twenty peptides were randomly selected for this analysis. For a single-peptide analysis, only tools that accepted peptide inputs were used, thus excluding MEGAN, MPA, and ProPHAnE. As a baseline, BLAST2GO v5.2.5 [23] was used to retrieve GO terms from a peptide. This tool was chosen due to its ability to obtain direct GO annotation via a universal algorithm, BLAST (Basic Local Alignment Search Tool). BLAST-P was used against the NCBI (nr) database with default parameters (no taxonomy filter, 1000 expectation value, 50 blast hits, word size of three with a low complexity filter enabled, 33 high-scoring segment pair (HSP) length cutoff, 0 HSP-hit coverage with GO mapping against the 2020_03 Goa version). Sensitivity and specificity of functional annotation differed between tools, in that some tools contained more GO terms than those found via BLAST2GO against the NCBI non-redundant (nr) database. Of the tools that used peptide-level input information (see Fig 1), Unipept was most similar to BLAST2GO outputs. Thus, eggNOG-mapper and MetaGOmics, which generally contained many more terms than BLAST/Unipept were scrutinized. To account for these extra terms, the hierarchical structure of GO terms was considered. GO terms assume a top-down hierarchical structure (directed acyclic graphs) which can be navigated using terms such as child, parent, ancestor and descendant. For any GO term, its ancestors consist of any number of less-specialized terms in its hierarchy. Conversely, its descendants consist of more-specific terms that would continue beneath its hierarchy. Parents and children are direct ancestors and direct descendants, respectively [20]. As an example, we can examine the ancestors and descendants of ion binding (GO:0043167): *metal ion binding* (descendant) → *cation binding* (child) → *ion binding* → *binding* (parent) → *molecular function* (ancestor).

To retrieve the ancestors and descendants of GO terms, the Python library GOATOOLS was used (20). Any ancestors, descendants, and ancestors' children were identified in the extra

GO terms found in eggNOG-mapper and MetaGOmics for each peptide. Terms that were uncategorized were labeled as 'extraneous'.

## Results

### General characteristics of the six tested software

**Input.** The six software tools that were evaluated in this study have been used for meta-proteomics analysis before (Fig 1). The inputs for each of these tools are different. EggNOG-mapper and Unipept take in only a peptide list and use established databases for annotation (eggNOG and UniProtKB databases, respectively). In contrast, search databases are required for MEGAN, MetaGOmics, MPA, and ProPHAnE which take in peptides with BLAST-P results, peptides with spectral counts, spectral search files, and spectral search results (from MPA, for example), respectively. MetaGOmics, MPA, and Unipept use UniProtKB as a data-base for annotation, while MEGAN requires results from the NCBI (nr) database. ProPHAnE can use EggNOG, PFAMs, TIGRFams databases and also custom databases for functional annotation.

**Analysis level.** The software tools differ in their level of analysis–some perform analysis at the peptide level (MetaGOmics, eggNOG-mapper and Unipept) and others perform analysis at the protein or protein-group level (MEGAN, MPA, and ProPHAnE).

**Outputs.** The software tools also generate variable outputs such as proteins annotated with GO terms (eggNOG-mapper, MEGAN, MetaGOmics, MPA, and Unipept 4.0), Inter-Pro2GO (MEGAN), EggNOG orthologous groups (eggNOG-mapper, MEGAN, and Pro-PHAnE), EC numbers (Unipept 4.0), and Pfam/TIGRFAM accessions (ProPHAnE). Given the variety of inputs, annotation databases, levels of analysis and output types, some variability in results can be expected if the same dataset is processed using these different software tools. To test the degree of variability, we used the published oral dysbiosis microbiome metaproteo-mics dataset (sample 737) [15].

### Variation in the number of GO term outputs from functional tools

Peptide reports for the sample grown without sucrose (NS) had 27,420 peptides (56,809 PSMs) and the sample grown with sucrose (WS) had 26,638 peptides (53,205 PSMs). Peptide search results from the oral dysbiosis dataset pair (see Methods) were processed to provide appropri-ate inputs for functional analysis. The outputs from the data processing through these software tools (Table 1) show that the total number of functional annotation groups differed for each software tool. For example, an orthologous group-based tool (e.g., MEGAN) can have as few as 1,665 groups while a tool with UniProt protein IDs (e.g., MPA) can have as many as 23,169 groups.

To facilitate comparison, the functional annotation groups were standardized into GO terms. Here, the number of GO terms derived from the tools is listed with and without dupli-cates. These two numbers are the same for tools that aggregate each GO term individually (MetaGOmics and Unipept). For tools that had GO terms translated, the unique (without duplicates) numbers are comparable with one another, ranging from as low as 1,056 (for MPA) to 6,155 (for eggNOG-mapper). The raw (with duplicates) numbers are less compara-tive but demonstrate the differences between tools that aggregate GO terms and those that do not. Since this study focuses on functional analysis, we filtered the GO terms to retain molecu-lar function GO terms. The number of unique molecular function GO terms ranged from as low as 634 (for MPA) to 1,613 (for MEGAN).

When compared with the numbers for all ontology GO terms, these molecular function GO terms number show their proportionality. For example, MPA had the highest molecular

Table 1. Functional analysis of the oral dysbiosis dataset using molecular function GO terms.

| Tool | EggNOG mapper | MEGAN | MetaGOmics | MPA | ProPHAnE | Unipept |
|---|---|---|---|---|---|---|
| Type of functional annotation | eggNOG orthologous groups | eggNOG orthologous groups | GO terms | Proteins | Protein families, eggNOG orthologous groups | GO terms |
| Total number of annotation groups | 18,440 | 1,665 | 2,829 | 23,169 | 3,999 | 3,471 |
| Total (and unique) number of translated GO terms for all ontologies | 533,066 (6,155) | 76,529 (4,155) | 2,829 (2,829) | 77,204 (1,056) | 189,054 (2,598) | 3,471 (3,471) |
| Total (and unique) number of translated molecular function GO terms | 88,582 (1,411) | 21,212 (1,613) | 900 (900) | 42,084 (634) | 57,208 (1,057) | 1,726 (1,726) |
| Total number of molecular function GO terms exclusive to the tool | 265 | 168 | 113 | 1 | 16 | 394 |
| Total number of molecular function expanded GO terms | 1,466 | 1,693 | 1,002 | 974 | 1,135 | 2,249 |
| Total number of molecular function expanded GO terms exclusive to the tool | 204 | 118 | 80 | 1 | 11 | 447 |

function composition at 60% (634 / 1,056) while eggNOG-mapper had the lowest at 23% (1,411 / 6,155). When comparing these molecular function GO terms between each tool, terms that were exclusive to a tool were highly present in Unipept at 22% (394 / 1,726), which contrasts with MPA, which only had one exclusive term: molybdopterin-synthase adenylyltransferase activity (GO:0061605).

To get a sense of the granularity of each tool's molecular function GO set, the numbers of each dataset's metaQuantome-expanded molecular function GO set is reported. metaQuantome expands each GO's hierarchy to list all possible ancestors. To avoid overloading 1 TB of RAM, the large datasets of eggNOG-mapper and ProPHAnE were collapsed by grouping via GO term and adding intensities. When compared with the numbers for molecular function GO terms, Unipept is found to contain the most expanded ancestors at 523 (2,249 minus 1,726) followed by MPA at 340 (974 minus 634). Proportionally, however, MPA had ancestors at 35% (340 / 974) of the expanded dataset and Unipept at 23% (523 / 2,249). The other tools had ancestor coverage at or less than 10% of their expanded GO sets: 3.8%, 4.7%, 10%, and 6.9% for eggNOG-mapper, MEGAN, MetaGOmics, and ProPHAnE, respectively.

Expansion to more ancestral terms affects how many terms were exclusive to each tool. For the expanded GO term sets, eggNOG-mapper, MEGAN, and MetaGOmics had around 50 fewer exclusive terms each when compared to their unexpanded counterparts. For ProPHAnE, there were only five fewer terms. In contrast to the other tools, MPA retained its only exclusive term, and Unipept had an increase of 50 exclusive terms through expansion.

The same tables were created for different category combinations as well: all GO terms (molecular function, biological process, and cellular component), biological process GO terms only, and cellular component GO terms only (**Supplement S1, Tables 1–3** in S1 File).

## Overlap analysis

Pairwise overlap analysis was performed on unique translated molecular function GO terms (sizes indicated in Table 1). The tool pairings that had the most overlap were MEGAN within ProPHAnE (92% of ProPHAnE's 474 terms are also in MEGAN's 1,613 terms) and Unipept within MPA (99% of MPA's 634 terms were found in Unipept's 1,726). Both observations are expected since ProPHAnE had a smaller output set and both MPA and Unipept use the UniProtKB database. Generally, tools with larger GO sets (e.g., MEGAN at 1,613) had better coverage in other tools' GO sets, as indicated by each row. However, even against a tool with a large GO set like MEGAN, 20 to 30% of most tools' annotations lacked overlap as a best case

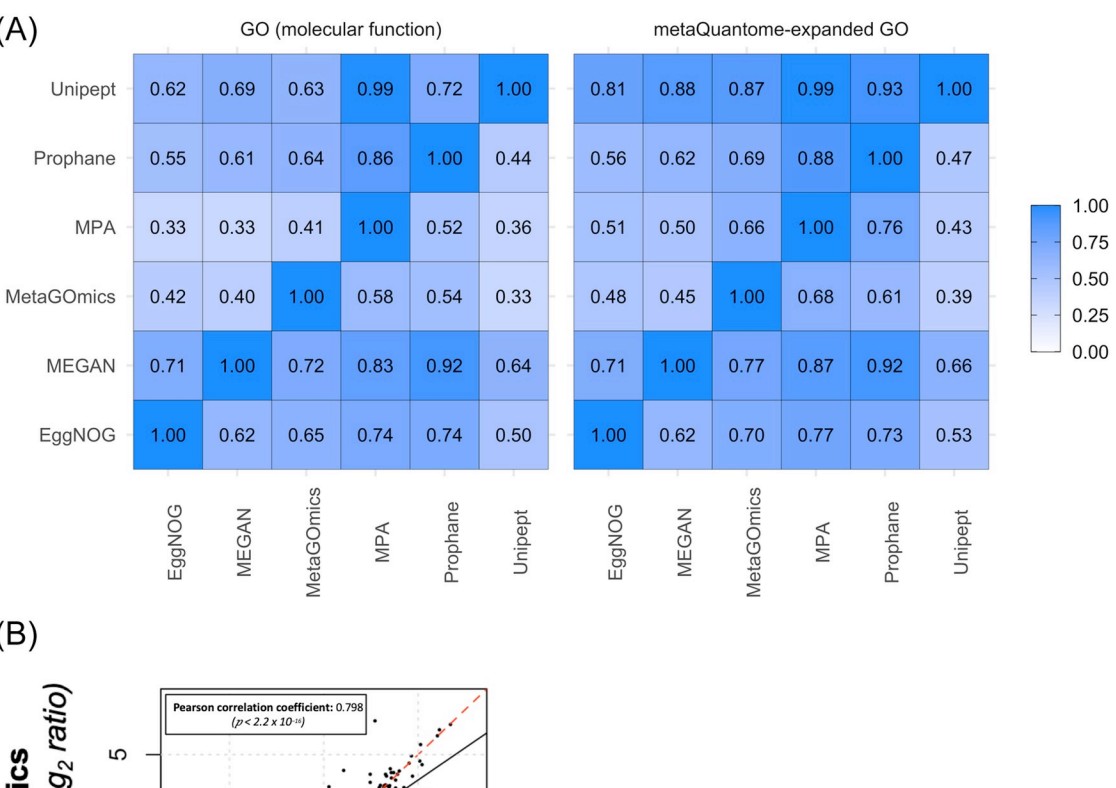

**Fig 2.** A) Qualitative and quantitative comparison of functional tools. Overlap of unique molecular function GO terms (left) and expanded GO terms (right) were compared amongst the six functional tools. Values were calculated as a fraction of the size of the term intersection (between the tools labeled on the column and row) over the total term size of the tool listed on the horizontal axis (column). Each functional analysis software tool was compared against each other. For example, for molecular function GO terms (left panel), 90% of the unique MPA term set is present in Unipept's unique term set. For molecular function expanded GO terms, the overlap is much larger for all tools within Unipept (right panel's top row). B) Comparison of quantitative expression for molecular function GO terms from Unipept and MetaGOmics. Log2ratio of spectral counts 'with sugar sample' (WS) against 'no sugar sample' (NS) was calculated for MetaGOmics- and Unipept-generated molecular function GO terms. Unipept identified 1,109 molecular function GO terms, while MetaGOmics identified 900 molecular function GO terms. The data points in the figure represent quantitative values for 460 molecular function GO terms that overlapped between Unipept and MetaGOmics.

scenario (Fig 2A left panel). In the metaQuantome-expanded versions of these GO sets, there is an overall improvement, notably with tools that had the most expansion (MPA and Unipept). Relative to GO term comparison, expanded GO term comparison shows an improvement due to an expanded representation of each term (Fig 2A right panel). The same overlap analysis was performed for all GO term ontologies, biological process GO terms, and cellular

component GO terms (Supplement S2 **Figures 2–4** in S1 File). Moreover, we also provide an Edwards-Venn diagram comparison of molecular function GO terms with unfiltered and expanded data (See **Supplement S2 Figures 1a and 1b** in S1 File) to provide overlap based on absolute numbers.

## Quantitative comparative analysis

After assessing the overlap of functional annotations (Fig 2A), we looked at the quantitative changes in GO terms for MetaGOmics and Unipept (Fig 2B). As mentioned earlier, MetaGO-mics and Unipept both generate GO terms as their native output. Comparison of quantitative expression using spectral counts for GO terms from Unipept and MetaGOmics was performed after normalization of spectral counts. Quantitative values of the overlapping molecular func-tion GO terms were represented (Fig 2B). The Pearson coefficient of this quantitative compari-son was found to be 0.798 with a significant p-value. Given that this is a quantitative comparison of the same dataset, a better quantitative correlation for overlapping molecular function GO terms was expected amongst two functional tools which used the same annota-tion database (UniProtKB).

Quantitative comparison for all GO terms, biological processes GO terms, and cellular component GO terms are available as **Supplement S3** in S1 File. The Pearson correlation for all of these comparisons shows lower values than for molecular function category (Supplement S3 **Figures 1–3** in S1 File).

## Differentially expressed GO terms across functional tools

A comparison of differentially expressed GO terms was performed by looking at the top five upregulated and top five downregulated terms for Unipept. Unipept was chosen since it aggre-gates on a GO term level and has more coverage at these extremes than MetaGOmics (the analogous table is available in **Supplement S4 Table 5** in S1 File). Overall, the fold changes for these top terms were somewhat maintained across most tools. Exceptions included eggNOG-mapper and MPA, which had either little to no coverage for some GO terms. Similar tables for the other tools can be found in **Supplement S4 3–7** in S1 File. We also provide expanded and filtered outputs for Unipept top five upregulated and down-regulated terms (**Supplement S4 1–2** in S1 File).

## Single peptide analysis

To determine the source of lack of overlap and quantitative correlation, we took a deeper look at the functional annotation of the top down regulated acetyl-CoA C-acetyltransferase activity (Fig 3 and **Supplement S5** in S1 File) and randomly selected peptides from the oral dysbiosis dataset (**Supplement S6** in S1 File). This term was chosen rather than the top-upregulated since the peptides associated with the term for each tool had no overlap between all peptide-level tools.

For example, a closer look at the peptides associated with acetyl-CoA C-acetyltransferase activity showed that for acetyl-CoA C-acetyltransferase, eggNOG-mapper, MetaGOmics, and Unipept assigned 53, 69, and 73 peptides, respectively. (Table 2). It should be noted that the PSM ratios from the sucrose-to-control dataset remained similar for Unipept, MetaGOmics and MEGAN. In contrast, eggNOG-mapper could assign only one peptide from the control dataset and could not assign any peptides from the sucrose dataset.

We performed analysis on single peptides that annotated acetyl-CoA C-acetyltransferase, which was the most down-regulated GO term using Unipept (Fig 3 and **Supplement S5** in S1 File). Analysis of the peptide that was assigned by EggNOG mapper (Fig 3) showed that

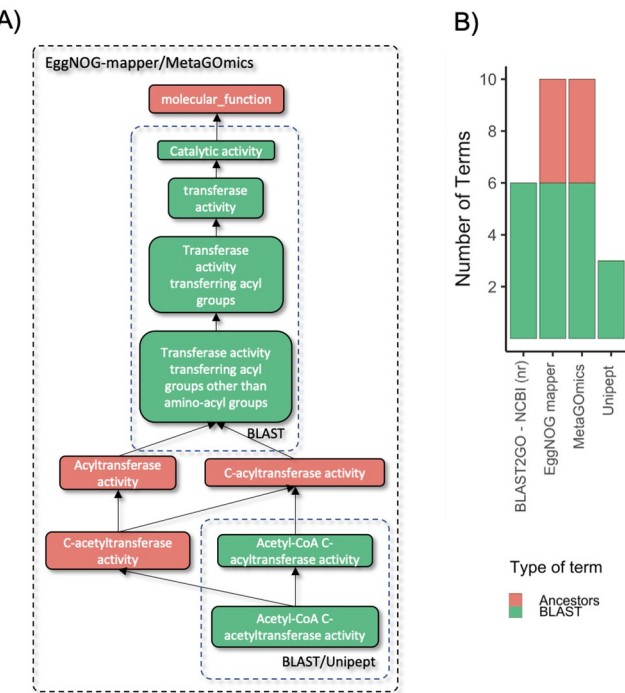

**Fig 3. Analysis of peptides associated with acetyl-CoA C-acetyltransferase activity.** A.) A combined GO hierarchy of unique terms annotated from a single peptide (sequence = FKDEIVPVVIPNK) for peptide-level tools (Unipept, MetaGOmics, and EggNOG-mapper), and a baseline tool: BLAST2GO—NCBI (nr). This peptide was selected from a group of 20 peptides randomly selected from all possible peptides that annotated 'Acetyl-CoA C-acetyltransferase' from the peptide-level tools. The peptide was selected since it shared results with the most number of peptides (10). Similar analyses for these other peptides are included in **Supplement S5** in S1 File. In this hierarchy, an arrow indicates "is a" or parent/children relationships. Colored blocks represent GO terms (color represents relationship type). Dashed block outlines (non-colored) are labeled with tools that encapsulate GO terms that were annotated by that tool. B.) A stacked bar chart representation of the related terms (descendants, ancestors, or ancestor's children). Colors correspond to these relation types. Green represents terms that were found through BLAST2GO's annotation via NCBI (nr). The other colors represent the relationship of the other terms in other tools to those BLAST GO terms. These types are quantified and stacked on one another to show the contributions of each relationship type to the overall GO hierarchy.

MetaGOmics and eggNOG-mapper annotations had related terms (ancestors). Unipept, on the other hand, identified GO terms that were more specific. MetaGOmics assigned ancestor and descendant terms for the rest of the peptides. This was in contrast to Unipept which identified specific GO terms—similar to the baseline BLAST searches against the NCBI nr database.

Bar diagrams for randomly selected peptides from the oral dysbiosis dataset (**Supplement S6** in S1 File) also show the number of ancestors' direct children, ancestor terms when using eggNOG-mapper, and MetaGOmics. It is clear that as compared to Unipept (and BLAST), these two functional tools (eggNOG-mapper and MetaGOmics) output more functional terms. The single-peptide analysis (Fig 3, **Supplement S5** and Supplement **S6 in** S1 File) was carried out for Unipept, MetaGOmics and eggNOG-mapper since they provide GO term annotations at a peptide level. As a result, we do not have results for the protein-level tools MEGAN, MPA and ProPHAnE.

In an effort to include more tools than just peptide-level tools, a single GO term, Xanthine dehydrogenase activity (XDA) (GO:0004854), was chosen for its annotation in all six tools. XDA also had fewer annotation groups in each tool, allowing for a manageable analysis. For

**Table 2. Comparison of the top five upregulated and top five downregulated molecular function GO terms of Unipept with the molecular function GO terms from the other tools from the oral dysbiosis dataset.**

| GO Term | Fold Change | Unipept | EggNOG | MEGAN | MetaGOmics | MPA | Prophane |
|---|---|---|---|---|---|---|---|
| glucosyltransferase activity | FC (WS / NS) | 6.87 (116 / 0) | 2 (3 / 0) | 1.46 (21 / 7) | 2.71 (117 / 17) | 3.01 (7.03 / 0) | 0.52 (0.44 / 0) |
| | Percentile (%) | 0.058 | 12.267 | 18.438 | 8.024 | 1.157 | 1.9 |
| dextransucrase activity | FC (WS / NS) | 6.67 (101 / 0) | - | - | 6.71 (104 / 0) | 3.01 (7.03 / 0) | - |
| | Percentile (%) | 0.086 | - | - | 0.035 | 1.157 | - |
| pyruvate oxidase activity | FC (WS / NS) | 6.38 (82 / 0) | - | - | 6.13 (69 / 0) | - | -0.09 (0 / 0.07) |
| | Percentile (%) | 0.115 | - | - | 0.071 | - | 57.689 |
| glyceraldehyde-3-phosphate dehydrogenase (NADP+) (non-phosphorylating) activity | FC (WS / NS) | 6.04 (65 / 0) | - | - | 6.02 (64 / 0) | - | - |
| | Percentile (%) | 0.144 | - | - | 0.141 | - | - |
| fructuronate reductase activity | FC (WS / NS) | 5.7 (51 / 0) | - | 0 (1 / 1) | - | - | - |
| | Percentile (%) | 0.173 | - | 52.072 | - | - | - |
| CoA-transferase activity | FC (WS / NS) | -7.84 (2 / 685.56) | -6.28 (0 / 76.8) | -8.56 (0 / 377) | -5.15 (15 / 568) | -4.52 (0 / 22) | -1.3 (0 / 1.47) |
| | Percentile (%) | 99.885 | 99.881 | 99.94 | 97.702 | 99.236 | 99.925 |
| acetyl-CoA C-acyltransferase activity | FC (WS / NS) | -7.92 (0 / 240.7) | -6.45 (0 / 86.16) | -8.59 (0 / 384) | -8.37 (0 / 330) | -5 (0 / 31) | -1.09 (0 / 1.13) |
| | Percentile (%) | 99.914 | 99.913 | 100 | 99.894 | 99.819 | 99.825 |
| butyrate-acetoacetate CoA-transferase activity | FC (WS / NS) | -8.36 (0 / 326.86) | - | - | -6.85 (0 / 114) | -4.52 (0 / 22) | -0.42 (0 / 0.33) |
| | Percentile (%) | 99.942 | - | - | 99.434 | 99.258 | 96.674 |
| glutaconate CoA-transferase activity | FC (WS / NS) | -8.41 (0 / 339.03) | - | - | -8.13 (0 / 279) | -4 (0 / 15) | -0.55 (0 / 0.47) |
| | Percentile (%) | 99.971 | - | - | 99.823 | 98.412 | 98.25 |
| acetyl-CoA C-acetyltransferase activity | FC (WS / NS) | -8.54 (0 / 369.94) | -6.45 (0 / 86.16) | -8.59 (0 / 384) | -8.37 (0 / 330) | -5.46 (0 / 43) | -1.09 (0 / 1.13) |
| | Percentile (%) | 100 | 99.913 | 100 | 99.859 | 99.987 | 99.825 |

Fold changes are featured here (descending for Unipept). For other tools, if there are multiple GO terms that match the top term, then the term with the highest absolute fold change is displayed. Additionally, spectral counts are indicated for "with sucrose" and "no sucrose" (WS / NS) conditions which are used to calculate the displayed fold change (FC) $= \log_2\left(\frac{WS+1}{NS+1}\right)$. Percentiles are included to indicate the position of that particular term in that GO set containing all ontologies (0 = most upregulated; 100 = most downregulated).

each tool, any GO terms that accompanied XDA on the same annotation group (i.e., peptide, protein, protein family, and eggNOG orthologous group) were organized into a GO hierarchy. The level of coverage and expansion is visually represented in two components similar to Fig 3.

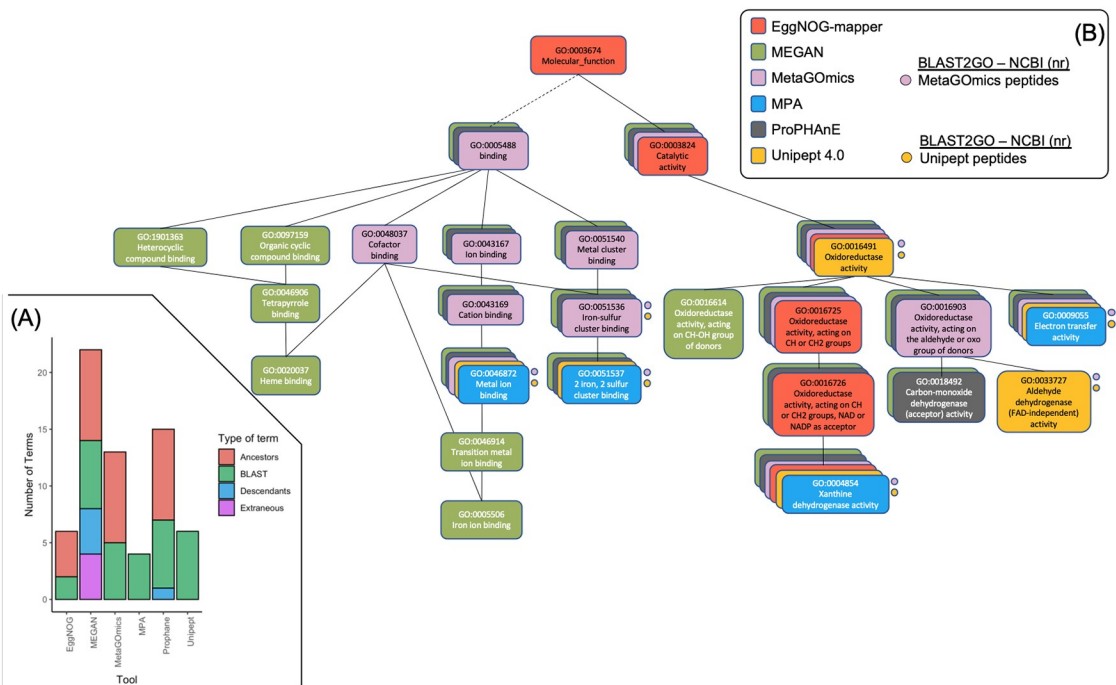

**Fig 4. Gene ontology hierarchy analysis of a single GO term for all six tools.** A) A stacked bar chart representation of the related terms (descendants, ancestors, ancestor's children, or extraneous) for xanthine dehydrogenase activity (XDA) for all six functional tools. Colors correspond to these relation types. Green represents terms that were found through BLAST2GO's annotation via NCBI (nr). The other colors represent the relationship of the other terms in other tools to those BLAST GO terms. These types are quantified and stacked on one another to show the contributions of each relationship type to the overall GO hierarchy. B) A single GO hierarchy analysis of xanthine dehydrogenase activity (XDA) (GO:0004854) for all six functional tools. For each tool, the GO terms of annotation groups containing XDA are arranged in a GO hierarchy for each tool. Each GO term is contained in a rounded rectangle. All six hierarchies are layered upon one another (with the largest in the background and the smallest in the foreground). Each color represents a tool (not a GO relationship). Solid lines represent "is a" or parent/child relationships and the dashed line indicates a connection that is present in the overall combined hierarchy, but not in any of the individual hierarchies. Circles indicate GO terms derived from BLAST2GO—NCBI (nr) results using peptides from the peptide-level tools (MetaGOmics, Unipept, EggNOG-mapper). GO terms not represented by more than one peptide are omitted (hence why eggNOG-mapper is not represented here).

The first component (Fig 4A) is a stacked bar chart similar to the one created for single peptide analysis (Fig 3). Here, available peptides from EggNOG-mapper (one peptide), MetaGOmics (12 peptides), and Unipept (52 peptides) that annotated XDA were used as input for BLAST2GO through the NCBI (nr) database. Amongst the 53 unique peptides, we retained 50 peptides that had GO sets with more than one peptide that supported that annotation. These 50 peptides were annotated by BLAST2GO and the GO terms were collated and duplicates were removed. Like Fig 3, the hierarchy relationships from the six functional tools' GO terms to BLAST2GO terms are listed in Fig 4A. The expansiveness of tools that output orthologous groups is highlighted here (such as MEGAN and ProPHAnE) which include both descendant and ancestor terms relative to BLAST2GO. MetaGOmics, a peptide-level functional tool, has an expanded hierarchy, but does not have descendants, relative to BLAST2GO. EggNOG-mapper and Unipept have the same amount of terms but eggNOG-mapper has a focused yet expanded hierarchy while Unipept has terms on the same level, similar to BLAST2GO. MPA is similar to Unipept, but with fewer terms.

To visualize the distributions in Fig 4A, the GO relationship hierarchies of each tool were constructed and overlaid on another in Fig 4B. Here, the anchor points (BLAST2GO terms) were labeled with circles. Each tool's level of expansiveness and specificity are visualized here.

For eggNOG-mapper, the GO terms annotated on the XDA-containing single protein (339724.XP_001539262.1 Xanthine dehydrogenase) have a hierarchy that is highly specific and expanded all the way. For MEGAN, there is high coverage and expansion from only two COGs: "COG1529 Aldehyde oxidase and xanthine dehydrogenase, molybdopterin binding" and "COG4631 xanthine dehydrogenase". For MetaGOmics, there are 12 peptides that annotate XDA and 13 associated GO terms resulting in an expanded hierarchy with moderate coverage. For MPA, there were two proteins ("W3Y6K1 selenium-dependent xanthine dehydrogenase" and "T0T10 xanthine dehydrogenase, molybdenum binding subunit") with four specific UniProt-derived GO terms. For ProPHAnE, there were two protein families that resulted in a hierarchy similar to MetaGOmics. For Unipept, there were 52 peptides containing nine unique molecular function GO terms. As a filtering mechanism, GO terms that were annotated in a number of proteins less than 5% of the total protein count per peptide were removed (GO:0016788 "hydrolase activity, acting on ester bonds", GO:0008270 "zinc ion binding", GO:0009056 "catabolic process"). Only six GO terms remained in a less expanded hierarchy similar to MPA. After listing out these hierarchies, the visual differences between functional tool annotations is apparent.

## Discussion

Using metaproteomics to investigate microbiomes has gained importance primarily due to its ability to identify functional roles of diverse taxonomic groups within a complex microbial community. In order to evaluate the software tools that are available for functional analysis of metaproteomics dataset, we used a published dataset of oral dysbiosis from plaque samples derived from dental caries prone children. In this study, the effect of sucrose on a dental plaque community was assessed wherein principal components analysis showed that the functional content exhibited better separation of the sucrose-treated samples from control samples as compared to taxonomic profiles [15]. Although, not a ground truth dataset, we chose this published dataset since it was thoroughly investigated. It is important to note that while taxonomy based ground-truth datasets are available [24], generating a functional microbiome ground truth dataset is not a trivial exercise.

For all of the functional studies, it is important that the software tools used offer results that facilitate a sound biological interpretation. Functional analysis tools either use peptide-level inputs (eggNOG-mapper, MetaGOmics and Unipept) or through other means (MEGAN, MPA and ProPHAnE). Functional annotations are generally performed using functional databases such as Gene Ontology (GO) [20] and Kyoto Encyclopedia of Genes and Genomes (KEGG) [25], or using databases that catalogue the evolutionary relationships of proteins such as orthologous groups (eggNOG). However, all of these databases/approaches are affected by issues associated with annotation quality [26].

Peptides or proteins can also be searched against annotated databases such as NCBI (nr) database [27] and UniProtKB by using algorithms such as BLAST-P or DIAMOND. For taxonomic assignments, peptides unique to a taxonomic unit are used to identify the taxonomic unit [28], while the majority of peptides are assigned at a higher taxonomic level (such as kingdom, phylum, etc.) and cannot be used to identify a lower taxonomic level i.e., strain, species, or genus. In contrast, for functional analysis the identified peptides are assigned to a protein and then a function. This offers an advantage of using functional information of peptides that have conserved function across taxa, even though they can only be assigned to relatively high taxonomic levels such as kingdom, phylum, etc. However, protein assignments and functional GO term assignments are hierarchical thus making it difficult to assign them to a single

function. Moreover, this is confounded by the issue that peptides/proteins can be assigned to multiple functional GO terms.

All of these issues are observed in our results. For example, the database size and composition and underlying algorithm used is reflected in the number of GO terms detected for each software at molecular function GO terms level (Table 1). The results from the overlap indices plot (Fig 2) is also noteworthy, though not surprising considering the variability in database and underlying algorithms used to annotate functions. As expected, the overlap index improves when using expanded GO terms to remove the variability between GO hierarchies (as demonstrated as well in Fig 4). We were surprised to see that the quantitative correlation between molecular function terms identified by MetaGOmics and Unipept was not better than 0.798, especially since this correlation was based on the overlapping GO terms. The non-overlap of GO terms (and hence quantitative values) can be explained by the hierarchical nature of GO terms—wherein the same peptides might have been assigned GO terms at varying levels. For example, the MetaGOmics algorithm expands its own hierarchy while Unipept does not (as demonstrated in Figs 3 and 4). For overlapping GO terms, the discrepancy in quantitative values might be explained by considering that peptides might have been assigned to different GO terms by different algorithms. For example, through the single GO analysis of 'Xanthine Dehydrogenase Activity' (XDA), only 12 peptides in MetaGOmics annotated XDA while 40 additional (52 total) peptides in Unipept annotated XDA (Fig 4).

When the ranking of differentially expressed molecular function GO terms was considered, we found that most tools were in agreement when estimating fold changes. However, there were minor discrepancies in a few of these up-regulated molecular function GO terms (Table 2). This indicated that some of the peptides were assigned to variable GO terms or not assigned at all by some software algorithms. In the case of peptide deformylase activity, which was the top-ranked differentially expressed molecular function GO term using Unipept, we observed that most of the peptides were not assigned using eggNOG-mapper. This may be attributed to the fact that eggNOG-mapper uses fine-grained orthologues to focus on novel sequences. This would result in the loss of quantitative information during eggNOG-mapper analysis, given that only one peptide in eggNOG-mapper annotated 'Peptide deformylase activity'. This peptide was only in NS, which goes against the expected value calculated from more WS counts. The other peptide-level tool, MetaGOmics, had a similar amount of spectral counts to Unipept when examining 'Peptide deformylase activity'. Surprisingly, a protein-level tool, MEGAN, had similar values as well. MPA, did not have similar spectral count values for a few of the upregulated terms, but did have relatively similar fold change calculations for the fourth and fifth upregulated terms and downregulated terms as well.

A closer look at the peptides assigned in the study showed the various terms and hierarchy at which molecular function GO terms were assigned (Fig 3 and **Supplement S5** and **S6** in S1 File). This shows the need for consideration of GO terms detected especially when ancestor, descendants, and any extraneous terms were detected. When this analysis was scaled up to include all tools that annotated xanthine dehydrogenase activity (Fig 4), the differences in annotation were still apparent. One thing to note, however, is that some of these terms were translated from other groups. For example, within MEGAN, proteins were annotated with COGs. These COGs were then translated into a large and expanded hierarchy of GO terms. ProPHAnE contains even more modules for translation from multiple groups: TIGR-FAM2GO, Pfam2GO, and OG2GO (eggNOG orthologous groups). These translations from different types of functional annotation need to be considered when comparing between tools. This highlights the importance of expanding the hierarchy in this analysis through tools such as metaQuantome [16] which allows for differences to be attributed to the underlying algorithm and database rather than the GO hierarchy. Additionally, being able to sift through,

aggregate, and filter through the hierarchy is important for not only comparing between tools, but also using them in a metaproteomics study.

Given all of these observations, we noted a few variables that might need to be considered for a more effective functional analysis. For example, some of the software seems to generate outputs with additional GO information. This is also augmented by the hierarchical structure of GO terms and the issue of multiple GO terms for the same protein (Fig 4). Adding a filtering step after navigating the hierarchical structure of GO terms and reporting signal from noise will aid in arriving at fairly consistent outputs. The usage of a metaproteomics tool such as metaQuantome or an R Bioconductor package such as GO.db (10.18129/B9.bioc.GO.db); extensively used during this analysis, can help navigate these hierarchical structures.

Based on our knowledge, this is the first study that evaluates the performance of various functional tools in the field of metaproteomics. We acknowledge a few limitations in the study, including performing analysis on a single replicate paired-samples and using spectral counts for quantitative analysis. This design was used primarily to address the current capacities of various software that were compared. Most of the software evaluated in the study supports analysis of single-replicate, spectral-count based analysis. We are currently evaluating MS1-based precursor-intensity tools and DIA-based tools [29] for protein quantitation, which we hope will provide more accurate quantitative data for these analysis programs in the future.

While using the same post-search inputs from the same dataset (except for MPA, which performed the X! Tandem search on a sectioned database internally), we were anticipating slight variability in the outputs from the six tools—mainly since they used variable databases for functional annotation along with underlying variability in algorithms. However, we were surprised by the incomplete overlap in GO terms (Fig 2A), particularly their quantitative outputs (Fig 2B and Table 2). This was rather noteworthy for tools that generated native GO terms as outputs. A closer look at the peptides and their functional assignment showed that software tools such as Eggnog-mapper and MetaGOmics generated a lot of information that was related but not specific to the protein. Unipept on the other hand showed more narrow and specific functional information. As a result of these observations, it is our opinion that in order to measure the functional component of the microbiome, there is a need for further refinement in databases used for annotation and also the underlying algorithms and filtering of the outputs generated, so that relevant functional information can be easily parsed out from these software tools. There is an opportunity, for example, to improve on the underlying database. This will become increasingly important when evaluating microbiomes from environments where metagenome and metatranscriptome data for peptide spectral matching is available, but not functionally annotated. Metaproteomics researchers face the challenge of analyzing the function of such microbiome systems and software developers have a major role in tackling this challenge. In order to assign functions to such metagenomics data, functional assignment after taxonomic binning is being used. This annotated metagenomics data can be used as a template to assemble metatranscriptomics data and generate protein databases for metaproteomic analysis. We also highlight the importance of targeted metaproteomics analysis to validate peptides and proteins as a follow up to key functions identified in discovery metaproteomics research.

As a result of the study, we have implemented two functional analysis tools (Unipept and EggNOG Mapper) which use distinct approaches related to database searching and filtering of results for usage in the Galaxy environment (see **Supplement S7** in S1 File); additionally, the computational performance of both tools are included (see **Supplement S8** in S1 File). We hope that access to these software will encourage metaproteomics researchers to explore these tools either individually or within Galaxy workflows [7,16,21,30,31]. We also hope that this evaluation encourages software developers to develop tools that generate the right balance of

annotation and results processing so that the functional analysis, which is the essence of meta-proteomics research, can be explored with confidence.

## Supporting information

**S1 File.**
(DOCX)

## Acknowledgments

We would like to thank the European Galaxy team for the help in the support during Galaxy implementation. We would also like to thank Alessandro Tanca (Porto Conte Ricerche, Italy), Mak Saito and Noelle Held (Woods Hole Oceanographic Institute, Woods Hole, MA) for discussion during the functional tools analysis. We would like to thank Tim van den Bossche (Ghent University, Belgium) for valuable inputs and thank Emma Leith (University of Minnesota) for proofreading the manuscript.

We also acknowledge the support from the Minnesota Supercomputing Institute for maintenance of the Galaxy instances and supercomputing resources used for this analysis.

## Author Contributions

**Conceptualization:** Caleb Easterly, Pratik D. Jagtap.

**Data curation:** Subina Mehta, Praveen Kumar, Joel Rudney, Pratik D. Jagtap.

**Formal analysis:** Ray Sajulga, Caleb Easterly, Michael Riffle, Bart Mesuere, Subina Mehta, Praveen Kumar, James Johnson, Stephan Fuchs, Pratik D. Jagtap.

**Funding acquisition:** Timothy J. Griffin.

**Investigation:** Ray Sajulga, Caleb Easterly, Subina Mehta, Praveen Kumar, Pratik D. Jagtap.

**Methodology:** Caleb Easterly, Bart Mesuere, James Johnson, Pratik D. Jagtap.

**Project administration:** Pratik D. Jagtap.

**Resources:** Bjoern Andreas Gruening.

**Software:** Michael Riffle, Bart Mesuere, Thilo Muth, Henning Schiebenhoefer, Stephan Fuchs, Brook L. Nunn.

**Supervision:** Carolin A. Kolmeder, Pratik D. Jagtap.

**Validation:** Ray Sajulga.

**Writing – original draft:** Ray Sajulga, Caleb Easterly, Pratik D. Jagtap.

**Writing – review & editing:** Ray Sajulga, Michael Riffle, Bart Mesuere, Thilo Muth, Subina Mehta, Praveen Kumar, James Johnson, Bjoern Andreas Gruening, Henning Schiebenhoefer, Carolin A. Kolmeder, Stephan Fuchs, Brook L. Nunn, Joel Rudney, Timothy J. Griffin, Pratik D. Jagtap.

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
