## [Decision Letter · Decision Letter 0]

11 Mar 2020

PONE-D-20-04130

Survey of metaproteomics software tools for functional microbiome analysis.

PLOS ONE

Dear Dr Jagtap,

Thank you for submitting your manuscript to PLOS ONE. After careful consideration, we feel that it has merit but does not fully meet PLOS ONE’s publication criteria as it currently stands. Therefore, we invite you to submit a revised version of the manuscript that addresses the points raised during the review process.

The overall assessment of this work is positive but reviewers have raised several issues that need to be addressed. Regarding the comparison and presentation of the results, the choice of Galaxy should be better justified. Including the run time would also enhance the results. In-depth arguments are also required in particular regarding the discussion raised by reviewer1 on the use of GO terms. Careful accounting of the reviewers' comments can definitely make this paper a flagship for users of metaproteomics analysis pipelines.

We would appreciate receiving your revised manuscript by April 30, 2020. To enhance the reproducibility of your results, we recommend that if applicable you deposit your laboratory protocols in protocols.io, where a protocol can be assigned its own identifier (DOI) such that it can be cited independently in the future. For instructions see: http://journals.plos.org/plosone/s/submission-guidelines#loc-laboratory-protocols

We look forward to receiving your revised manuscript.

Kind regards,

Frederique Lisacek

Academic Editor

PLOS ONE

Journal Requirements:

2.Please ensure that you refer to Figure 4 in your text as, if accepted, production will need this reference to link the reader to the figure.

Reviewers' comments:

Reviewer's Responses to Questions

**Comments to the Author**

1. Is the manuscript technically sound, and do the data support the conclusions?

Reviewer #1: Partly

Reviewer #2: Yes

2. Has the statistical analysis been performed appropriately and rigorously? 

Reviewer #1: Yes

Reviewer #2: No

3. Have the authors made all data underlying the findings in their manuscript fully available?

Reviewer #1: No

Reviewer #2: No

4. Is the manuscript presented in an intelligible fashion and written in standard English?

Reviewer #1: Yes

Reviewer #2: Yes

5. Review Comments to the Author

Reviewer #1: The manuscript aims to benchmark six available metaproteome annotation tools, eggNOG-mapper, MEGAN6, MetaGOmics, MetaProteomeAnalyzer (MPA), ProPHAnE, and Unipept.

To benchmark the tools, an “oral dysbiosis” dataset was selected which includes the samples of plaque bacteria that were grown with and without sucrose. For both of these samples, the predicted peptides were normalised and mapped to functional groups/terms using the benchmarked tools. Subsequently the differential expression of the terms were calculated for each of the tool. The core of the benchmark is the comparison of these differentially expressed terms.

The manuscript also extensively compares inputs and outputs of the benchmarked tools. As they vary widely in that respect, the presented methodology generates specific inputs for each of the tools and aims to synthesise comparable output datasets.

The authors find that although the differentially expressed annotated terms are all significantly correlated between the tools, they exhibit a large degree of variability. At the end two of the tools (one which exhibits high specificity and sensitivity, and one that mapped the largest amount of terms) were then incorporated into the Galaxy platform.

Growing interest in and availability of metaproteomics data will result in significant need for annotation tools and established annotation pipelines. The presented manuscript could be of significant interest to metaproteomics community and have immediate impact on metaproteomics analysis methods. Therefore it is important to assure that the analysis presented in the manuscript is correct and interpretable.

The core aspect of the presented methodology is assigning the differential expression to each GO term and making the sets comparable between the tools. Unfortunately this is also the weakest part of the manuscript.

For the methods to be comparable, each GO term, for each peptide assigned by the method, should be projected to the root of its ontology and the differential expression should be calculated at each of the GO terms after that projection. It is not clear if this is done here. As such, without this projection, the methods that assign more granular terms will have diminished chance of picking up the signal, as the differential expression signal would be divided onto the term’s children whereas less granular method could incorporate the signal from all the term’s children. GO prediction benchmarking should reward the methods that assign more granular terms, as the GO terms could always be projected. Instead it seems that described method of the comparison effectively punishes for it. This is such a critical weakness, that lets me to believe I did not understand the methodology correctly. However the only part of the manuscript that describes the implementation of GO ontology traversal is found in the analysis of the 20 random peptides which is not related to the core of the comparison.

To partially mitigate the above point GO terms were translated into higher level GO slim subset, however this subset consists of only 147 terms, which may be too broad to generate the interpretable differential expression signal. I say potentially, as authors did not include in the manuscript any results from GO slim comparison except calculating Jaccard index between the tools’ outputs, which is not particularly informative.

All differentially expressed list of terms (projected to root) from all the tested methods, for all three GO ontologies should be included as a part of supplementary data. This will allow readers a way to judge the validity of the method by themselves. The overlap comparisons and “Top ten” terms for each method not sufficient for a meaningful interpretation.

Another point that should be addressed is a need for a biological interpretation of the results generated by the methods. There must be some processes that we should expect to see being differentially expressed between sugar rich and a sugar deprived environment. For example this may include the GO terms associated with energy metabolism, biofilm creation, growth etc. Expert curators with the knowledge of plaque bacteria could highlight few higher level GO terms that have a potential to be differentially expressed between the two conditions. If such term, selected by curators, does not exhibit metaproteome level differential expression, then we should not expect it to be picked up by any method by chance, conversely if at least one method finds such term (or its descendant) to be differentially expressed it probably won’t be by chance alone. The ranking of these curated term in the methods output could potentially highlight which of the tools is more suitable for finding biological signal in this particular metaproteome data. Of course it is important to note that the conclusions derived from this test cannot be trivially extended to all other biological datasets, nonetheless there is an insight to gain from assessing distinct aspects of the differences between the tools’ outputs.

Other issues:

The authors state in the abstract that the “A BLAST analysis against the Universal Protein Knowledgebase revealed that the sensitivity and specificity of functional annotation differed between tools”. Highlighting this measure in the abstract is debatable. Primarily because there is no reason why simple BLAST against UniProt would be regarded as gold standard. If it is, there is no need for more sophisticated tools. It’s a useful measure to some extent, but it’s not clear what that measure actually conveys as it seems to me that it depends heavily on the degree of overlap between UniProt and the peptide search database (HOMD in this case). In such case almost by definition Unipept would have to generate the most precise GO, as it is directly based on the peptide matches to UniProt database. This is also what we see.

Execution time of the tools, is always an important factor in the decision which of the available tools to use. For large scale analysis it is often the deciding factor. It would be helpful to point out the total computation time of each of the pipelines that were used to generate final GO terms. If a database only starts with peptides the Mascot search should be added to the computation time, but the execution time of the tool itself should also be highlighted.

The authors state that DIAMOND has “similar sensitivity to BLAST”. There is a price for the speed-up of search and it seems to be sensitivity [1, Fig. 2b]. In my experience this is especially true for a very diverged proteins (identity <30%), where DIAMOND significantly loses sensitivity. The exact numbers will depend on the testing methodology.

Figure 3. Legend should be corrected. Pink in “Type of term” legend relates to “Extraneous” terms, which are not present in the bar chart. However, in the same figure pink is used also as a color of eggNOG terms that are not part of the MetaGOmics output. This is misleading as only only one of the eggNOG terms (Formylmethionine deformylase activity”) could be considered as “extraneous” in relation to the gold terms.

[1] Steinegger, Martin, and Johannes Söding. "MMseqs2 enables sensitive protein sequence searching for the analysis of massive data sets." Nature biotechnology 35.11 (2017): 1026-1028.

Reviewer #2: This is an interesting manuscript comparing different tools used for metaproteomic data processing. The manuscript has some limitations that need to be addressed.

-This seems to be a combination of comparing tools and an add-on to the Galaxy platform. Sufficient details are provided for the comparison of the tools. However, it is not possible to review the add-on to the Galaxy platform as limited details are provided. It is a bit strange to have both in one manuscript. I would suggest to focus the manuscript on comparing the tools.

-Table 3. Comparing ranks is really not useful and somewhat misleading. Instead, the results from each GO term should be analyzed (# of peptide mapped and quantification) and the results statistically compared.

-Figure 4 should either be removed or moved as supplementary.

-The authors did a good job at highlighting the limitations of their analysis.

-A few typos throughout the manuscripts.

6. PLOS authors have the option to publish the peer review history of their article (what does this mean?). If published, this will include your full peer review and any attached files.

Reviewer #1: No

Reviewer #2: No

---

## [Author Response · Author response to Decision Letter 0]

26 Aug 2020

We would like to thank the editors of the journal to extend the submission deadline to September 7th 2020. We believe that the extended time and communication with multiple authors has helped us to improve on the quality of the manuscript.

Along with this cover letter we have the revised ‘Manuscript’ along with revised manuscript with edits and Response to Reviewers where we have answered questions by the reviewers.

We will greatly appreciate if Plos ONE can facilitate the review of this revised manuscript and make a decision on the suitability of its publication in your journal.

---

## [Decision Letter · Decision Letter 1]

14 Sep 2020

PONE-D-20-04130R1

Survey of metaproteomics software tools for functional microbiome analysis.

PLOS ONE

Dear Dr. Jagtap,

Thank you for submitting your manuscript to PLOS ONE. After careful consideration, we feel that it has merit but does not fully meet PLOS ONE’s publication criteria as it currently stands. Therefore, we invite you to submit a revised version of the manuscript that addresses the points raised during the review process.

Mainly, the remaining inconsistencies spotted by the reviewer should be attended to.

We look forward to receiving your revised manuscript.

Kind regards,

Frederique Lisacek

Academic Editor

PLOS ONE

Reviewers' comments:

Reviewer's Responses to Questions

**Comments to the Author**

1. If the authors have adequately addressed your comments raised in a previous round of review and you feel that this manuscript is now acceptable for publication, you may indicate that here to bypass the “Comments to the Author” section, enter your conflict of interest statement in the “Confidential to Editor” section, and submit your "Accept" recommendation.

Reviewer #3: (No Response)

2. Is the manuscript technically sound, and do the data support the conclusions?

Reviewer #3: Yes

3. Has the statistical analysis been performed appropriately and rigorously? 

Reviewer #3: Yes

4. Have the authors made all data underlying the findings in their manuscript fully available?

Reviewer #3: Yes

5. Is the manuscript presented in an intelligible fashion and written in standard English?

Reviewer #3: No

6. Review Comments to the Author

Reviewer #3: The authors describe in their manuscript a comparison of tools providing functional annotations, applied here on metaproteomics data. A preprocessed common dataset is used for all six tools. The functional annotation is investigated through directly predicted or secondarily obtained gene ontologies to have a common framework to do comparisons.

As clearly stated in the title and acknowledged by the authors, it is more a survey than a well-controlled benchmarking and it is limited to providing an overview of the available methods while demonstrating that their predictions vary substantially. The authors discuss the cause of these discrepancies as being for instance the variety of reference database, and advocate for the improvement and curation of these resources. Since the field lacks any benchmarking for that kind of aims, this study is a first step and therefore valuable.

While I appreciate the design of the different evaluations and the discussion, I find the overall description of the tools and their evaluation rather confusing, which in my opinion impair the understanding of the results if you are not a user of each of these six different pipelines. I am unsure whether this is caused by multiple errors in the manuscript or a lack of rigor in the systematic description of the inputs/outputs/databases. Different sections, figures, and tables seem to contradict or to consider the features of each tool differently. I would advocate for polishing and clarification. The language is clear.

- In general, page5:"Standard procedures were used for each tool" is not sufficient to identify when methods overlap regarding their data sources. Databases underlying the tool should be listed explicitly for each of the six methods, and not scattered across different figure and tables, no matter bundled with the tool or to be provided externally.

- For instance, eggNOG db is used by multiple tools, probably for different purpose. Figure 1 does not connect eggNOG db to MEGAN, while methods state page5:"the analysis was performed using eggNOG ". Table 2 states that eggNOG obtain functional annotation of type "proteins" while MEGAN obtain them from eggNOG groups. All together, this prevents the reader from understanding whether MEGAN and eggNOG get part of their functional annotation from a common source or not at all.

- Figure 1 is not that helpful as it does not reflect the difference between tools that matter during the evaluation. Especially, it does not clearly separate and define the databases used for classification of the input and their functional annotation. In some part of the manuscript, other db (page5: "For MetaGOmics, a list of peptides and the HOMD were uploaded (...)") are mentioned but not present on Figure 1.

- Two tools are described as natively predicting GO, vs four which do not (page8:"tools that natively output GO terms (i.e., Unipept and MetaGOmics)"). But on Figure 1, five out of six tools output GO, so this seems to be a different interpretation of natively predicting GO. Moreover, methods states that page6:"GO terms (...) are common annotation type throughout most of the tools (i.e., eggNOG-mapper and Unipept) (...) However, MEGAN, ProPHAnE, and MPA do not provide direct GO term outputs". Only five tools mentioned, most of the tools being two of them (!), and not matching previous statements or Figure1.

- I am unsure whether MPA uses the output of XTandem and Peptideshaker since the "input" folder in the supplementary archive contain a fasta that contains cRAP protein, which resemble a database use for search and filtering (277K_for_737_MPA_with_cRAP.fasta). I suppose this could indicate the actual input is the initial spectrum files identical to what is given to XTandem (in methods: page5:"For MPA, the same parameters as for SearchGUI were used"). This would contradict the statement in the discussion page21:"While using the same post-search inputs from the same dataset". Or there is something else missing (cRAP-filtereds spectrum files?) as input for MPA in the supplementary archive.

- To clarify all the points above, a visual summary of the workflow from the first difference appearing in any tool to the end of the process with the translation of GO would be beneficial, either as supplementary or as replacement of Figure 1.

- The definition of peptide-centric / peptide-level / protein-centric / protein-level could be clarified, if these terms are necessary to understand the input and outputs that lead to GO predictions attached to individual proteins, OG, or peptides. These terms seems to be used in contradictory ways and are confusing.

page9: "Of the tools that used peptide-level information (see Figure 1)". => Figure 1 contains "protein-level information" as possible output, and "peptides" as possible input. The reader could assume that "peptide-level information" are those without the checkmark for "protein-level information" outputs in Figure 1, but "used" does not mean "outputed".

Page9: "tools that provided peptide-level outputs were used, thus excluding MEGAN, MPA, and ProPHAnE" This does not match any of the possibilities depicted in Figure 1, but does match page10:"analysis level" in the results section.

page17: " protein-level tools is highlighted here (MEGAN and ProPHAnE) " Again, this is not the same list... the meaning might be "some of the protein-level tools" but the general confusion about these terms cast doubts.

page18: "a peptide-centric (eggNOG-mapper, MEGAN, MetaGOmics and Unipept) or protein-centric approach (MPA and ProPHAnE)." This matches the grouping in figure1 and seems correct, but "centric" does not tell whether to focus on inputs or outputs and MPA and ProPHAnE do not have the same input type according to figure 1.

- page16: "The single- peptide analysis (Figure 3, Supplement S5 and Supplement S6) was carried out for Unipept, MetaGOmics and eggNOG-mapper since they provide GO term annotations at a peptide level. As a result, we do not have results for the protein-level tools MPA and ProPHAnE." Related to previous comments, MEGAN is not included, no justification here... is it because of protein-level vs peptide-level? Or inhability to provide GO term ?

minor:

- the Jaccard Index is only mentioned in the discussion while it was removed elsewhere during refactoring. Please check whether it is still intended to refer to this or prefer something else to match figure 2

- MEGAN5 and MEGAN6 are mentioned as being the evaluated version. Please check whether this is a typo.

- Supplementary figure S5: one color is not labeled (pink)

7. PLOS authors have the option to publish the peer review history of their article (what does this mean?). If published, this will include your full peer review and any attached files.

Reviewer #3: No

---

## [Author Response · Author response to Decision Letter 1]

14 Oct 2020

Reviewer #3: The authors describe in their manuscript a comparison of tools providing functional annotations, applied here on metaproteomics data. A preprocessed common dataset is used for all six tools. The functional annotation is investigated through directly predicted or secondarily obtained gene ontologies to have a common framework to do comparisons.

As clearly stated in the title and acknowledged by the authors, it is more a survey than a well-controlled benchmarking and it is limited to providing an overview of the available methods while demonstrating that their predictions vary substantially. The authors discuss the cause of these discrepancies as being for instance the variety of reference database, and advocate for the improvement and curation of these resources. Since the field lacks any benchmarking for that kind of aims, this study is a first step and therefore valuable.

While I appreciate the design of the different evaluations and the discussion, I find the overall description of the tools and their evaluation rather confusing, which in my opinion impair the understanding of the results if you are not a user of each of these six different pipelines. I am unsure whether this is caused by multiple errors in the manuscript or a lack of rigor in the systematic description of the inputs/outputs/databases. Different sections, figures, and tables seem to contradict or to consider the features of each tool differently. I would advocate for polishing and clarification. The language is clear.

We appreciate the comments by the reviewer. In the revised version of the manuscript we have addressed the concerns raised by making changes to the Figure 1 and text changes that offer clearer description of the methodology.

- In general, page5:"Standard procedures were used for each tool" is not sufficient to identify when methods overlap regarding their data sources. Databases underlying the tool should be listed explicitly for each of the six methods, and not scattered across different figure and tables, no matter bundled with the tool or to be provided externally.

We have made changes to Figure 1 that has a “Database(s)” section that outlines which internal databases were used by each software tool. We hope that sets the stage for the reader to navigate the manuscript.

- For instance, eggNOG db is used by multiple tools, probably for different purpose. Figure 1 does not connect eggNOG db to MEGAN, while methods state page5:"the analysis was performed using eggNOG ". Table 2 states that eggNOG obtain functional annotation of type "proteins" while MEGAN obtain them from eggNOG groups. All together, this prevents the reader from understanding whether MEGAN and eggNOG get part of their functional annotation from a common source or not at all.

We thank the reviewer for pointing this out. The new “Database(s)” section in the revised Figure 1 now explicitly outlines which databases are used for each software tool, with particular emphasis on EggNOG and UniProtKB databases. In the updated Figure 1, MEGAN is now connected to the eggNOG database. 

Additionally, Table 1 (which the reviewer has referred to as “Table 2”) is revised from stating “proteins” to “eggNOG orthologous groups” as an annotation type, to demonstrate similarity with MEGAN.

- Figure 1 is not that helpful as it does not reflect the difference between tools that matter during the evaluation. Especially, it does not clearly separate and define the databases used for classification of the input and their functional annotation. In some part of the manuscript, other db (page5: "For MetaGOmics, a list of peptides and the HOMD were uploaded (...)") are mentioned but not present on Figure 1.

The issue is now addressed via the new “Database(s)” section in Figure 1. We have also added an input element - Human Oral Microbiome Database (HOMD) - that connects to MetaGOmics software so that there is consistency between the figure and description.

- Two tools are described as natively predicting GO, vs four which do not (page8:"tools that natively output GO terms (i.e., Unipept and MetaGOmics)"). But on Figure 1, five out of six tools output GO, so this seems to be a different interpretation of natively predicting GO. Moreover, methods states that page6:"GO terms (...) are common annotation type throughout most of the tools (i.e., eggNOG-mapper and Unipept) (...) However, MEGAN, ProPHAnE, and MPA do not provide direct GO term outputs". Only five tools mentioned, most of the tools being two of them (!), and not matching previous statements or Figure1.

We have addressed this by removing the misleading “GO terms” row in the table in Figure 1. Now, the revised figure, via the “Output” section, clearly illustrates native GO term outputs from Unipept and MetaGOmics.

- I am unsure whether MPA uses the output of XTandem and Peptideshaker since the "input" folder in the supplementary archive contain a fasta that contains cRAP protein, which resemble a database use for search and filtering (277K_for_737_MPA_with_cRAP.fasta). I suppose this could indicate the actual input is the initial spectrum files identical to what is given to XTandem (in methods: page5:"For MPA, the same parameters as for SearchGUI were used"). This would contradict the statement in the discussion page21:"While using the same post-search inputs from the same dataset". Or there is something else missing (cRAP-filtereds spectrum files?) as input for MPA in the supplementary archive.

The reviewer raises important questions on how MPA analysis was carried out. This was amended by specifying that MPA performs its own searches, using the same inputs (MGF files) and parameters as SearchGUI and requires a FASTA database that was generated using a database sectioning approach. The statement in the discussion (page 21) referred to by the reviewer was modified to convey this information.

- To clarify all the points above, a visual summary of the workflow from the first difference appearing in any tool to the end of the process with the translation of GO would be beneficial, either as supplementary or as replacement of Figure 1.

Figure 1 has now been revised to illustrate differences and similarities between tools more easily via a grid-like and color-coded design. Additionally, a dedicated section for “GO Term translation” is included within Figure 1.

- The definition of peptide-centric / peptide-level / protein-centric / protein-level could be clarified, if these terms are necessary to understand the input and outputs that lead to GO predictions attached to individual proteins, OG, or peptides. These terms seem to be used in contradictory ways and are confusing.

page9: "Of the tools that used peptide-level information (see Figure 1)". => Figure 1 contains "protein-level information" as possible output, and "peptides" as possible input. The reader could assume that "peptide-level information" are those without the checkmark for "protein-level information" outputs in Figure 1, but "used" does not mean "outputed".

This is addressed by including the word “input” in the sentence to clarify that peptide-level inputs were used for these tools. “Input” was used rather than “output” due to the clear peptide-level input information in Figure 1.

Page9: "tools that provided peptide-level outputs were used, thus excluding MEGAN, MPA, and ProPHAnE" This does not match any of the possibilities depicted in Figure 1, but does match page10:"analysis level" in the results section.

This is addressed by removing the “Protein level information” row in the table in Figure 1. The wording was also adjusted to center around “peptide inputs”.

page17: " protein-level tools is highlighted here (MEGAN and ProPHAnE) " Again, this is not the same list... the meaning might be "some of the protein-level tools" but the general confusion about these terms cast doubts.

The language here is changed to look at tools that “output orthologous groups”

page18: "a peptide-centric (eggNOG-mapper, MEGAN, MetaGOmics and Unipept) or protein-centric approach (MPA and ProPHAnE)." This matches the grouping in figure1 and seems correct, but "centric" does not tell whether to focus on inputs or outputs and MPA and ProPHAnE do not have the same input type according to figure 1.

Instead of using “peptide-centric” and “protein-centric”, the sentence was modified to divide the tools between those that use “peptide-level inputs” and those that use “other means”.

- page16: "The single- peptide analysis (Figure 3, Supplement S5 and Supplement S6) was carried out for Unipept, MetaGOmics and eggNOG-mapper since they provide GO term annotations at a peptide level. As a result, we do not have results for the protein-level tools MPA and ProPHAnE." Related to previous comments, MEGAN is not included, no justification here... is it because of protein-level vs peptide-level? Or inhability to provide GO term ?

MEGAN was not initially included, due to mislabeling on the previous Figure 1. Now, it has been added (and Figure 1 has been corrected).

minor:

- the Jaccard Index is only mentioned in the discussion while it was removed elsewhere during refactoring. Please check whether it is still intended to refer to this or prefer something else to match figure 2

The “Jaccard Index” was now correctly renamed to “overlap index”

- MEGAN5 and MEGAN6 are mentioned as being the evaluated version. Please check whether this is a typo.

This is a typo. MEGAN5 was used for this survey.

- Supplementary figure S5: one color is not labeled (pink)

Color has been added (pink) for the “Extraneous” category.

---

## [Editor Report · Decision Letter 2]

16 Oct 2020

Survey of metaproteomics software tools for functional microbiome analysis.

PONE-D-20-04130R2

Dear Dr. Jagtap,

We’re pleased to inform you that your manuscript has been judged scientifically suitable for publication and will be formally accepted for publication once it meets all outstanding technical requirements.

Kind regards,

Frederique Lisacek

Academic Editor

PLOS ONE
---

## [Editor Report · Acceptance letter]

29 Oct 2020

PONE-D-20-04130R2 

Survey of metaproteomics software tools for functional microbiome analysis 

Dear Dr. Jagtap:

I'm pleased to inform you that your manuscript has been deemed suitable for publication in PLOS ONE. Congratulations! Your manuscript is now with our production department. 

Kind regards, 

on behalf of

Dr. Frederique Lisacek 

Academic Editor

PLOS ONE